# Tree Species and Epiphyte Taxa Determine the “*Metabolomic niche*” of Canopy Suspended Soils in a Species-Rich Lowland Tropical Rainforest

**DOI:** 10.3390/metabo11110718

**Published:** 2021-10-21

**Authors:** Albert Gargallo-Garriga, Jordi Sardans, Abdulwahed Fahad Alrefaei, Karel Klem, Lucia Fuchslueger, Irene Ramírez-Rojas, Julian Donald, Celine Leroy, Leandro Van Langenhove, Erik Verbruggen, Ivan A. Janssens, Otmar Urban, Josep Peñuelas

**Affiliations:** 1Global Change Research Institute of the Czech Academy of Sciences, The Czech Academy of Sciences, Bělidla 986/4a, CZ-60300 Brno, Czech Republic; j.sardans@creaf.uab.cat (J.S.); klem.k@czechglobe.cz (K.K.); urban.o@czechglobe.cz (O.U.); 2Global Ecology Unit CREAF-CSIC-UAB, Consejo Superior de Investigaciones Científicas (CSIC), Bellaterra, 08193 Catalonia, Spain; Josep.Penuelas@uab.cat; 3CREAF, Cerdanyola del Vallès, 08193 Catalonia, Spain; 4Department of Zoology, College of Science, King Saud University, P.O. Box 2455, Riyadh 11451, Saudi Arabia; afrefaei@ksu.edu.sa; 5Centre of Microbiology and Environmental Systems Science, University of Vienna, Althanstrasse 14, 1090 Vienna, Austria; lucia.fuchslueger@uantwerpen.be; 6Department of Biology, University of Antwerp, BE-2610 Wilrijk, Belgium; irene.ramirezrojas@uantwerpen.be (I.R.-R.); leandro.vanlangenhove@uantwerpen.be (L.V.L.); erik.verbruggen@uantwerpen.be (E.V.); ivan.janssens@uantwerpen.be (I.A.J.); 7Centre for Ecology and Conservation, University of Exeter, Penryn TR10 9FE, UK; julian.donald@univ-tlse3.fr; 8AMAP, University Montpellier, CIRAD, CNRS, INRAE, IRD, 34000 Montpellier, France; celine.leroy@ird.fr; 9ECOFOG, CNRS, CIRAD, AgroParisTech, INRAE, Université des Antilles, Université de Guyane, 97310 Kourou, France

**Keywords:** bacteria, canopy soils, epiphyte, French Guiana, metabolomics

## Abstract

Tropical forests are biodiversity hotspots, but it is not well understood how this diversity is structured and maintained. One hypothesis rests on the generation of a range of metabolic niches, with varied composition, supporting a high species diversity. Characterizing soil metabolomes can reveal fine-scale differences in composition and potentially help explain variation across these habitats. In particular, little is known about canopy soils, which are unique habitats that are likely to be sources of additional biodiversity and biogeochemical cycling in tropical forests. We studied the effects of diverse tree species and epiphytes on soil metabolomic profiles of forest floor and canopy suspended soils in a French Guianese rainforest. We found that the metabolomic profiles of canopy suspended soils were distinct from those of forest floor soils, differing between epiphyte-associated and non-epiphyte suspended soils, and the metabolomic profiles of suspended soils varied with host tree species, regardless of association with epiphyte. Thus, tree species is a key driver of rainforest suspended soil metabolomics. We found greater abundance of metabolites in suspended soils, particularly in groups associated with plants, such as phenolic compounds, and with metabolic pathways related to amino acids, nucleotides, and energy metabolism, due to the greater relative proportion of tree and epiphyte organic material derived from litter and root exudates, indicating a strong legacy of parent biological material. Our study provides evidence for the role of tree and epiphyte species in canopy soil metabolomic composition and in maintaining the high levels of soil metabolome diversity in this tropical rainforest. It is likely that a wide array of canopy microsite-level environmental conditions, which reflect interactions between trees and epiphytes, increase the microscale diversity in suspended soil metabolomes.

## 1. Introduction

Tropical rainforests are one of the world’s most biodiverse [1,2,3], yet threatened ecosystems [4,5], in which the origin and maintenance of high levels of local-scale tree species coexistence [3,6] have been explained by several factors, including heterogeneity of levels of disturbance [1] and soil traits and nutrients [2,7,8,9,10,11,12], micro-site singularities [13], topographical features [14] (such as slope aspect and steepness [15], and long-term divergences in species-specific defenses against herbivores [16]. Rainforest canopies host large numbers of floral and faunal specialists with evolved adaptations to niches that are distinct from those elsewhere in the rainforest ecosystem [17]. However, rainforest canopy communities remain poorly studied, despite their critical role in global climate function and cycling of carbon and nitrogen [18]. Investigating soils suspended in the canopy (i.e., suspended soils, epiphyte-associated soils) is particularly challenging, but it is believed that wide gradients in environmental conditions have led to the development, through biological adaptation processes, of distinct micro-habitats comprising characteristic species [19,20,21] to create within-ecosystem biodiversity hotspots. For example, although tropical rainforest suspended soils sustain a complex and diverse faunal trophic food web structure similar to that of forest floor soils [22], trophic positions tend to be occupied by contrasting species [23,24] that further boost the overall high levels of biodiversity of these ecosystems [25].

Suspended soils along tree trunks and branches retain nutrients and water essential for the development of epiphytic plants [26] that directly depend on these limited pools of available nutrients [27]. These plants are morphologically and physiologically adapted to facilitate the accumulation of leaf litter and water and to maximize atmospheric and invertebrate-mediated delivery of nutrients [15,28,29] in the physically harsh and variable environmental conditions that prevail within the canopy [30]. Epiphytes represent up to 35% of the vascular floral diversity of wet tropical forests [31,32], and studies of these plants have tended to focus on water and nutrient uptake strategies [33]. Epiphytes have evolved numerous remarkable adaptations to facilitate nutrient uptake. For example, certain members of the Bromeliaceae family form water and litter-storing phytotelmata and take up nutrients through leaf-absorbing trichomes [29,34,35], whilst *Asplenium* ferns intercept falling leaf litter, which is then stored as organic matter adjacent to the roots [27,36]. For both epiphytes, this accumulation of leaf litter leads to the formation of organic matter or epiphyte-associated soils that harbor characteristic levels of microbial diversity [37,38]. The extent of this harboring is not well understood, in part due to the difficulty of sampling this habitat, but also due to our inability to characterize the entire spectrum of the metabolome. Little is known about variation in the chemical composition of the organic matter in these suspended and epiphyte-associated soils, or about their association with canopy position or tree species [39].

The study of complex chemical and physiological traits of suspended soils is challenging, given that well established analytical methods tend to focus on individual or specific groups of compounds [40], such as chemical defensive molecules acting as toxins and deterrents [41]. The plant metabolome comprises a complex set of primary (sugars, amino acids, and nucleotides) and secondary metabolites (terpenoids and phenolics) that are synthesized by plants in response to specific environmental conditions [42]. The analysis of the whole soil metabolome has been demonstrated to reflect metabolic function across soil microbial, floral, and faunal communities [43] and may be used as a proxy for soil community composition and function [44,45]. Thus, the development of highly sensitive ecometabolomic techniques enables us to study metabolic responses to dynamic shifts and gradients in environmental conditions at a range of scales, from the plant to the ecosystem [40,46,47].

Tree diversity has been shown to influence soil heterogeneity in tropical forests [48], whereby the impacts of differences in litter species composition and their released leachate shapes both forest floor traits [49,50] and suspended canopy soils [24]. Thus, changes in soil traits and contrasting microbe community composition, nutrient status, and enzyme content of forest floor and suspended soils have been observed to depend on tree species [26,38,51]. Canopy soils have higher concentrations of organic matter than ground soils [17,52]. A recent study in a tropical mountain rainforest in Costa Rica revealed that canopy soils harbor very different symbiotic and fungi communities compared to ground soils and have much more enzymatic activity [26]. Here, we use metabolomic analyses to test the hypotheses that (i) tropical forest epiphyte-associated soil metabolome profiles vary depending on the epiphyte taxa, (ii) suspended soil and epiphyte-associated soil metabolomes differ in their composition, and (iii) these differences are shaped by tree species creating a wide array of distinct metabolome niches. We expect that canopy soil metabolomes contrast with those of the forest floor, and that the differences will depend on the organic matter acquisition strategies of epiphytes or of microbes, within-canopy niche position, microclimate conditions [53], and host tree species characteristics [24].

## 2. Results

Soil metabolomes differed as a result of their sample location (forest floor versus canopy) and their association with other plants (host tree and epiphyte species) (Table 1 and Table 2). These findings were revealed by their separation along the first two axes of the corresponding PLSDA biplots for soil type (Figure 1), tree species (Figure 2), and epiphyte-associated and suspended soils (Figure 3 and Figure 4), accounting for 24, 10, and 13% of the variation in metabolome, respectively. Overall, an abundance of 2496 of the 2757 metabolite signals were higher in suspended soils than in forest floor soils, particularly for groups that include phenolic, aliphatic, and polycyclic aromatic compounds; unsaturated fatty acids; terpenes; most sugars; organic and amino acids associated with the Krebs cycle (succinic, lactic, malic, citric, chlorogenic, pyruvic, jasmonic, and abscisic acids); and those with nitrogenous bases (adenosine, guanosine, guanine, thymine, cytosine, and cytidine). The concentrations of 690 compounds were greater in soils associated with epiphytes than those without (Figure 1). Analysis of the metabolome of suspended soils and epiphyte-associated soils showed separation of that of Bromeliaceae and Orchidaceae along axes 1 and 2, respectively (Figure 3).

### Enrichment of Metabolic Pathways

There was metabolic pathway enrichment particularly for those pathways related to amino acids and nucleotides, in suspended soils compared to forest floor soils (some > 8 times) (Appendix A) and in epiphyte-associated soils compared to suspended soils (Figure 5 and Figure 6).

## 3. Discussion

### 3.1. Metabolome of Forest Floor and Canopy Soils

The metabolome of forest floor soils was distinct from that of canopy soils, including both with and without epiphytes. Organic matter decomposition rates in canopy soils contrast with those on the floor; processes tend to be slower in the canopy [54].

Our results showed that an abundance of several plant metabolite compounds, such as primary metabolites and phenolic groups, were greater in canopy soils than in forest floor soils, indicating a greater proportion of compounds of plant origin. This is consistent with previous studies observing that the organic matter contents in floor soil (~6% C) greatly differ with those in canopy soil or suspended soil (~35% C) [52]. We found higher abundances of metabolites such as amino acids, nucleotides, and compounds related to energy metabolism in suspended soils compared to forest floor soils, indicating the role of tree leaf litter in the formation of suspended soils. Similarly, in a Costa Rican rainforest, higher concentrations of host tree compounds and nutrients were reported in canopy soils compared with forest floor soils, where there was also variation in the two soil types among tree species [24]. As expected, we found more organic compounds resulting from the metabolization of the leaves in canopy soils than in the forest floor soils, which contain much less organic carbon. This is again consistent with the origin of soil in canopies, which is mostly a result of plant litter, in contrast with the ground soil, which is mostly a result of bedrock weathering and leaching [55].

Our results showed a clear effect of tree species on canopy soil metabolomic profiles, regardless of association with epiphytes, indicating that species-specific environmental conditions in the canopy influence canopy soil processes. While suspended soils are directly influenced by the host tree through inputs of its leaf litter [56], variations in host tree species epiphyte communities with environmental conditions indicate that microsite factors may be key drivers of epiphyte community composition, rather than limitation of dispersal [57]. Given that epiphytes capture water and nutrients from the atmosphere and intercepted host tree leaf litter [58], variation in canopy soil metabolomes among host tree species and between suspended soils and those on the floor forest is expected, along with differences in key factors, such as soil C and nutrient concentrations [22]. For example, in a Hawaiian rainforest, a greater content of P in tree bark was found to be positively related to the presence and colonization of certain types of epiphytes [59], and concentrations of available nutrients in epiphyte-associated soils were shown to depend on tree host species [25]. Microscale changes in forest physicochemical traits are linked with tree species distribution in tropical forests [7,13,60] and canopy microclimate conditions, which vary with tree species architecture and vertical position in the forest, influencing epiphyte species composition and associated soil formation [61,62]. For example, rainforest tree canopies that intercept and accumulate higher amounts of water from fog were found to be positively related to greater epiphyte community biomass and diversity in both French Guiana [63] and Costa Rica [64].

We found tree species differences in forest floor and canopy soil metabolomic profiles, supporting studies that have shown tree species composition and diversity affect epiphyte species composition and function, due to the effects on soil nutrient concentration and availability [62,65], soil microbiota and invertebrate community composition [66], phosphorus concentration and availability [10,67,68], physical characteristics [69], fertility [70], and moisture and organic carbon content [71,72,73].

### 3.2. Metabolic Profile Differences among Canopy Soils

In addition to the effects of tree species on forest floor soil metabolomics, we also observed epiphyte taxon differences in metabolic profiles of epiphyte-associated soils, where there was greater upregulation of pathways compared to suspended soils. Epiphyte-associated soils receive organic compounds, such as epiphyte root exudates [56], that likely differ in chemical composition and function, leading to variation in soil metabolome among epiphyte taxa.

Epiphyte species composition and canopy accumulation of organic matter are a positive feedback of appropriate microclimate conditions, such as lower temperature and greater levels of humidity [74], that create environmental niches occupied by organisms, which may not otherwise persist [53,75,76]. For example, variation in levels of light penetration and moisture among canopy strata have been linked with contrasting epiphyte communities and function [77,78,79,80,81], supporting our finding that suspended soil metabolic profiles are associated with epiphyte taxa, while a study exploring microbe communities of suspended soils using artificial soil microcosms found a negative relationship between the volume of suspended soil and microbe diversity [36].

The differing modes of nutrient acquisition (roots versus leaf-absorbing trichomes) and variation in microclimate among epiphytes (wet in bromeliads versus dry in *Asplenium*) likely result in distinct microbial communities [82], as supported by the distinct metabolic profiles of bromeliad-associated soils. *Asplenium* ferns, which retain greater levels of moisture by their fibrous roots than suspended soils, are likely to provide an intermediate microclimate, as supported by our analysis that shows its metabolome differs from those associated with the other epiphyte taxa. Suspended soil microbial community composition is sensitive to changes in microclimate [38], and our study indicates that epiphyte-mediated influences on the levels of moisture in suspended soils and accumulation of nutrients may contribute to the associated microfauna community composition. We recommend future studies focus on the dynamics and drivers of these poorly understood epiphyte-associated soils.

Our analysis shows the influences of tree species and epiphyte taxa on variance in suspended soil metabolomic profiles. We found that soil metabolic profiles associated with *Asplenium* varied with tree species, indicating likely differences in associated environmental conditions and distinct local metabolomes.

### 3.3. Global Overview of the High Diversity of Metabolomic Profile in the Studied Soils

Currently, there is no clear consensus on the direction of effects of tree species on soil traits. It has been argued that local-scale soil properties cause the high levels of tree diversity in tropical forests [7,11,12,83]. However, our results clearly demonstrate that rainforest tree and epiphyte taxa influence heterogeneity and metabolic function of forest floor and canopy soils, with a positive feedback on their diversity. The dependence on forest floor soil traits of tree species in tropical forests [26,38,48,51] has been linked to differences in tree species leaf litter leachates [49,50]. Yet, our study is the first to show evidence of a general link between forest floor and canopy soil traits (here, metabolomics) and plant species composition in a tropical forest. Our results demonstrate that the greater abundance of metabolites related to plant species (particularly those related to biological production and expression, such as amino acids and nucleotides) and to plant stress-tolerance (such as phenolics) have a positive feedback effect on soil trait diversity and the key role of plant diversity in the maintenance of micro-scale soil biological and functional diversity in this rainforest in French Guiana.

## 4. Materials and Methods

### 4.1. Study Area

The study was conducted in September 2017 at the Pararé research station in Nouragues Nature Reserve (4°02′ N, 52°41′ W) in French Guiana, where lowland wet tropical forest represents 97% of total land cover [84]. A pronounced dry season, which extends from September to November, is associated with the displacement of the inter-tropical convergence zone; the mean annual temperature is 25.8 °C, with an annual amplitude of 2 °C (daily amplitudes of 7 and 10 °C in the rainy and dry seasons, respectively) [85,86]. The study area comprises an inselberg granite outcrop, with primary lowland rainforest of high floral diversity [87,88]. The Pararé research station and the adjacent 4-hectare research plot experience an average daily temperature and annual rainfall of 26 °C and 2861 mm, respectively [89]. Soils are classified as nutrient poor Acrisols [90]. Samples were collected within a fully inventoried 1.5-hectare plot at the Pararé Research Station, where canopy height ranges between 35 and 55 m [91].

### 4.2. Soil Sampling and Analysis

Thirteen species (Table 3) of trees across a range of families within the study plots were randomly selected at the Pararé Research Station, based on the presence of epiphytes and the feasibility of climbing. Using ropes to climb the trees, we collected specimens of *Asplenium* (*Asplenium serratum*), Bromeliaceae (*Aechmea aquilega* and *Mezobromelia pleiosticha*), Araceae (*Anthurium* sp. and *Philodendron* sp.), and Orchidaceae epiphytes from branches, recording their height and orientation in the canopy. The epiphyte-associated and suspended soils were removed from supporting branches, placed into sealed bags, and transported to the field camp. We sampled at least three epiphytes and three organic soils in each individual tree. Organic matter was removed from the roots of Araceae and Orchidaceae, the tanks of bromeliads, and the litter-trapping leaves of *Asplenium*. The soils were processed following the approach described by Gargallo-Garriga et al. [92], where samples were passed through a 0.5 × 0.5 cm mesh and placed in a paper bag, for temporary storage at −80 °C on dry ice and additional liquid nitrogen, prior to transfer to the EcoFoG lab at Kourou, where they were freeze dried for 48 h (Alpha 1-2 LD, Christ, Osterode am Harz, Germany). Then, the samples were ground to a fine powder and homogenized, using a pestle and mortar, and stored at −80 °C prior to analysis. In addition, we collected three replicates of soil (forest floor soil) around the individual tree, always in the same position and less than 1 m of distance from the trunk base.

### 4.3. Metabolite Extraction and Analysis

Following the approach described by Gargallo-Garriga et al. [93], the soil samples were extracted using a 1:1 methanol:H_2_O solution. Then, the extracted fraction was analyzed twice, by using the positive and negative ion modes of a liquid chromatography–mass spectrometer (LC-MS; UltiMate 3000 chromatographic system coupled to a LTQ Orbitrap XL high-resolution mass spectrometer, ThermoFisher Scientific, Waltham, MA, USA) that was equipped with a heated electrospray ionization source to perform metabolomic profiling. Samples were analyzed in MS and MS/MS under the same conditions, by selecting the top three parent ions of each scan. The raw data from the LC-MS were processed and compared using the XCMS 72 platform, as described in Gargallo-Garriga et al. (2020). We quantified the metabolites, after the elimination of peaks that were not consistently representative, based on the presence of mass/RT in at least three samples of any provenance, when each sample was then split and randomly queued into two LC-MS run batches. The peak areas corresponding to each metabolite were normalized based on the total peak areas in the sample. Neutral masses obtained in positive and negative modes were evaluated to avoid duplicates (same retention time and neutral mass in the different modes), retaining the most intense peaks. LC-MS chromatograms were obtained with a Dionex Ultimate 3000 HPLC system (Thermo Fisher Scientific/Dionex RSLC, Dionex, Waltham, MA, USA) coupled to an LTQ Orbitrap XL high-resolution mass spectrometer (Thermo Fisher Scientific, Waltham, MA, USA) equipped with an HESI II (heated electrospray ionization) source. Chromatography was performed on a reversed-phase C18 Hypersil gold column (150 × 2.1 mm, 3-µ particle size; Thermo Scientific, Waltham, MA, USA) at 30 °C. The mobile phases consisted of acetonitrile (A) and water (0.1% acetic acid) (B). Both mobile phases were filtered and degassed for 10 min in an ultrasonic bath prior to use. The elution gradient, at a flow rate of 0.3 mL per minute, began at 10% A (90% B) and was maintained for 5 min, then to 10% B (90% A) for the next 20 min. The initial proportions (10% A and 90% B) were gradually recovered over the next 5 min, and the column was then washed and stabilized for 5 min before the next sample was injected. The injection volume of the samples was 5 µL. HESI was used for MS detection. All samples were injected twice: once with the ESI operating in negative ionization mode (−H) and once in positive ionization mode (+H). The Orbitrap mass spectrometer was operated in FTMS (Fourier Transform Mass Spectrometry) full-scan mode with a mass range of 50–1000 m/z and high-mass resolution (60, 000). The resolution and sensitivity of the spectrometer were monitored by injecting a standard of caffeine after every 10 samples, and the resolution was further monitored with lock masses (phthalates). Blank samples were also analyzed during the sequence. The assignment of the metabolites was based on the standards, with the retention time and mass of the assigned metabolites in both positive and negative ionization modes (Appendix A).

### 4.4. Metabolite Identification and Quantification

We identified and determined metabolites based on comparison with our standard compound library (>200 compounds) and searching for each MS and MS/MS in KEGG and MASSBANK databases, following the approach described by Gargallo-Garriga et al. (2020). For those that could not be positively identified, we made tentative identifications based on MS/MS spectra as annotated metabolites. As a result, we assigned the stochiometry of 2757 metabolic compounds, and based on the stoichiometric ratios of nitrogen and hydrogen to carbon, and of oxygen to carbon [94,95], we classified the compounds into four broad families according to plant secondary metabolism, comprising highly-unsaturated polyphenols (192 compounds), polycyclic aromatics (142 compounds), aliphatics (1351 compounds), and phenolic compounds (280 compounds).

### 4.5. Statistical Analysis

We used permutational analysis of variance (PERMANOVA; Anderson 2001), using the ‘ADONIS’ function from the ‘vegan’ package [96] in R (R Development Core Team 2015), first to test for differences in soil metabolome profile in function of tree species and types of soil (forest floor soil, canopy without and with epiphytes) as independent factors (13 tree species) and a second PERMANOVA was conducted only with canopy suspendedsoils with tree species and soil with different epiphyte taxa (*Asplenium*, Bromeliaceae, Araceae, and Orchidaceae) as independent factors, in pursuit of the “metabolomic niche hypothesis” [97]. PERMANOVA determines variation within a distance matrix, which is assigned a priori groups for each level of independent factors, and compares the observed community matrix to a nonparametric null distribution, which here was based on 2000 permutations of the observed matrix.

We used partial least squares discriminant analysis (PLSDA) to characterize and visualize relations among soil metabolomes by position (forest floor and canopy soil, including suspended and epiphyte-associated soils) and the association with epiphyte taxa. We then conducted an enrichment pathway analysis (MetaboAnalyst 5.0, www.metaboanalyst.ca, accessed on 23 September 2021) to test for up- and downregulated metabolic pathways in comparing suspended soils without epiphytes with the suspended soils with different epiphyte taxa. This analysis allowed us to detect the metabolic pathways that are differently expressed between two sets of samples by comparing their metabolomics profiles.

## 5. Conclusions

An abundance of metabolites tended to be greater in canopy soils than in forest floor soils, and in epiphyte-associated soils than in suspended soils, where groups such as phenolics of plant origin and those involved in primary metabolomic pathways, such as those related to amino acid, nucleotides, and energy metabolism, were enriched. Tree species was a main driver of forest floor and canopy soil (both suspended and epiphyte-associated) metabolomic profiles.

## Figures and Tables

**Figure 1 metabolites-11-00718-f001:**
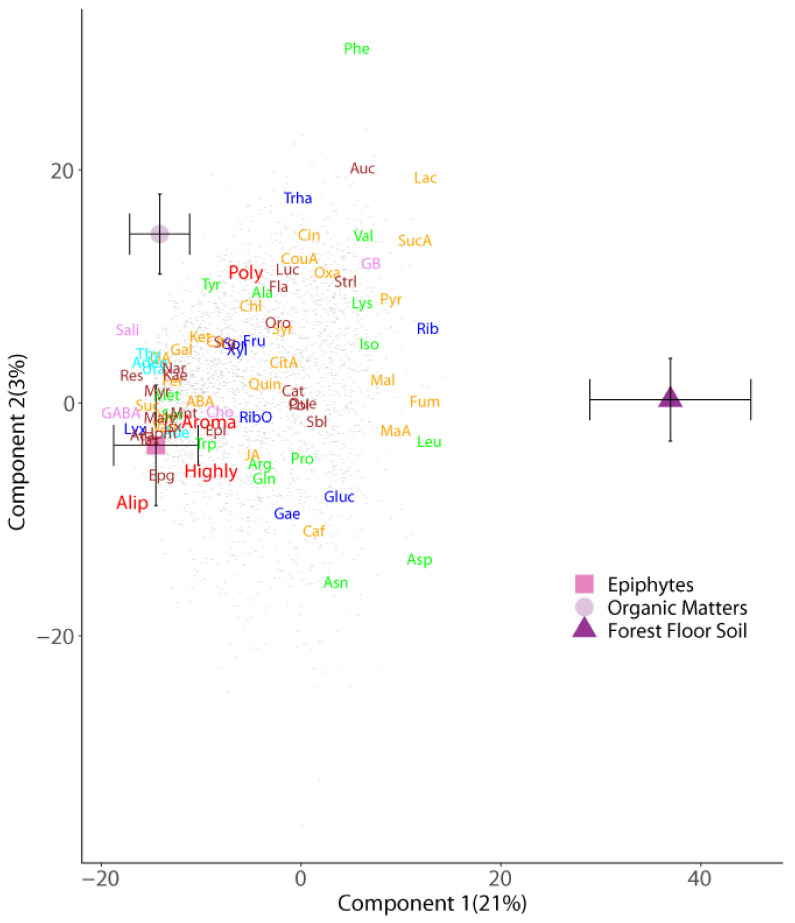
Partial least squares discriminant analysis biplots of metabolomic profiles of forest floor soils and epiphyte-associated and non-associated suspended soils (organic matter). Depicted values are means (±95% CI). Samples’ categorized scores (mean ± S.E.) by different soil types: “Epiphytes” (suspended soils associated with an epiphyte), “Organic Matters” (canopy soil without an epiphyte), and “Forest Floor Soil”. Soils are indicated by different colors: pink square, epiphytes; lilac circle, organic matters; purple triangle, forest floor soil. Loadings of the metabolites; only those with loadings >0.5 are depicted. The various metabolomic families are represented by different colors: dark blue, sugars; green, amino acids; orange, compounds involved in the metabolism of amino acids and sugars; cyan, nucleotides; brown, phenolics; red, others. Metabolites: arginine (Arg), asparagine (Asn), aspartic acid (Asp), glutamic acid (Glu), glutamine (Gln), isoleucine (Ile), lysine (Lys), leucine (Leu), methionine (Met), phenylalanine (Phe), serine (Ser), tryptophan (Trp), threonine (Thr), tyrosine (Tyr), valine (Val), adenine (Ade), adenosine (Ado), thymidine (TdR), chlorogenic acid (CGA), trans-caffeic acid (CafA), α-ketoglutaric acid (KG), citric acid (Cit), L-malic acid (Mal), lactic acid (Lac), abscisic acid (Abs), pyruvate (Pyr), succinic acid (SAD), pantothenic acid hemicalcium salt (Pan), jasmonic acid (JA), 5,7-dihydroxy-3,4,5–trimethoxyflavone (Fla), acacetin (AC), epicatechin (EC), epigallocatechin (EGC), homoorientin (Hom), isovitexin (Ivx), kaempferol (Kae), myricetin (Myr), quercetin (Qct), resveratrol (Rvt), saponarin (Sp), catechin hydrate (Cat), 3-coumaric acid (CouA), gallic acid (GA), quinic acid (QuiA), sodium salicylate (Sal), syringic acid (Syr), trans-ferulic acid (Fer), vanillic acid (Van), 2-deoxy-D-ribose (Rib), D-(-)-lyxose (Lyx), D-(+)-sorbose (Sor), D-(+)-trehalose dehydrate (Tre), aucubin (Auc). All the other small grey dots correspond to non-identified compounds.

**Figure 2 metabolites-11-00718-f002:**
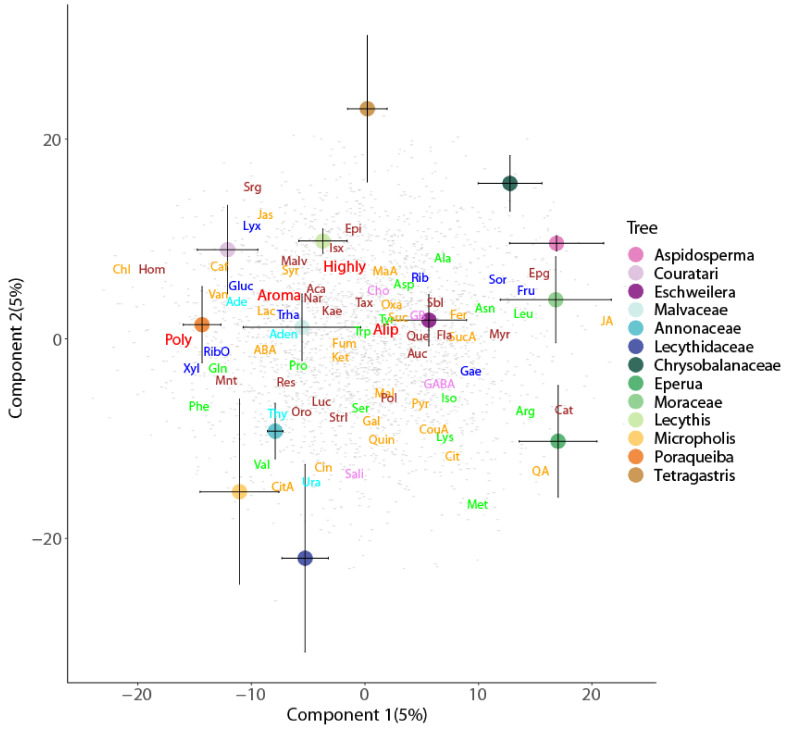
Partial least squares discriminant analysis biplots of metabolomic profiles of forest floor and suspended soils across 13 species of tree (Table 2). Depicted values are means (±95% CI). Metabolites as in Figure 1.

**Figure 3 metabolites-11-00718-f003:**
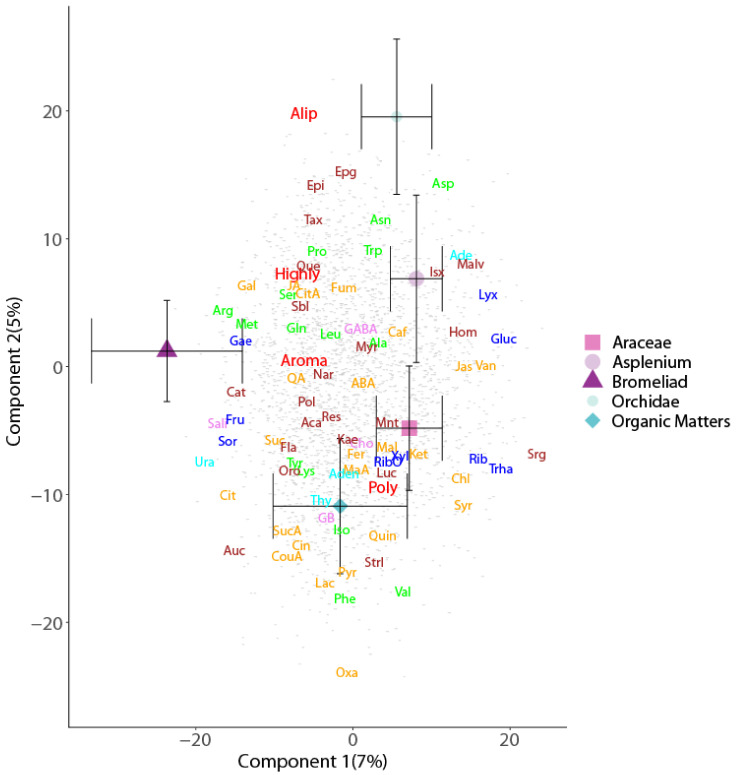
Partial least squares discriminant analysis biplots of metabolomic profiles of epiphyte (Araceae, *Asplenium*, Bromeliaceae, Orchidae)-associated and non-associated (organic matter) suspended soils. Depicted values are means (±95% CI). Metabolites as in Figure 1.

**Figure 4 metabolites-11-00718-f004:**
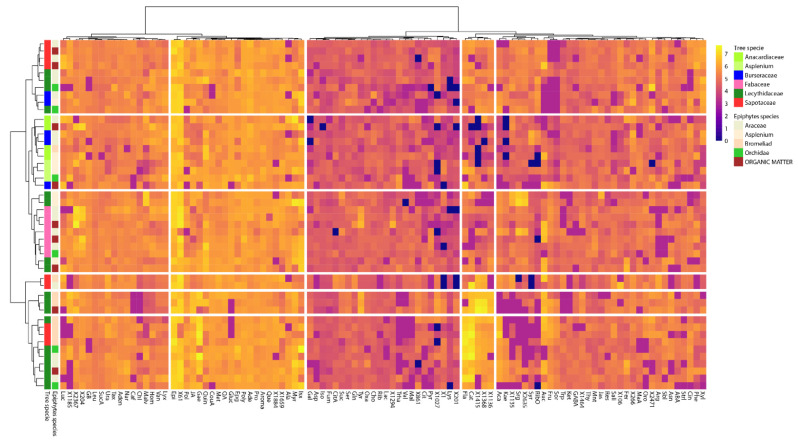
Heatmap clustering analysis of the metabolites identified in the foliar metabolomes of tree species and epiphyte species. Families and species are generally clustered together with the corresponding metabolites. The samples are represented horizontally, and the metabolites are represented vertically. Abundance is represented by the intensity of the color of each metabolite, with blue representing low abundance and yellow representing high abundance. The 100 most important metabolites that were differentially affected (*p* < 0.05) between species and family are represented.

**Figure 5 metabolites-11-00718-f005:**
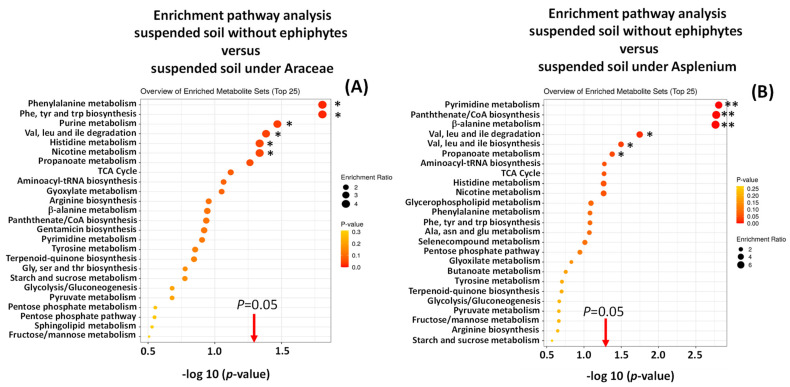
Enrichment pathway analysis of metabolomic profiles of suspended soils associated with Araceae (**A**) and Asplenium (**B**) compared with non-epiphyte soils. * *p* < 0.05, ** *p* < 0.01.

**Figure 6 metabolites-11-00718-f006:**
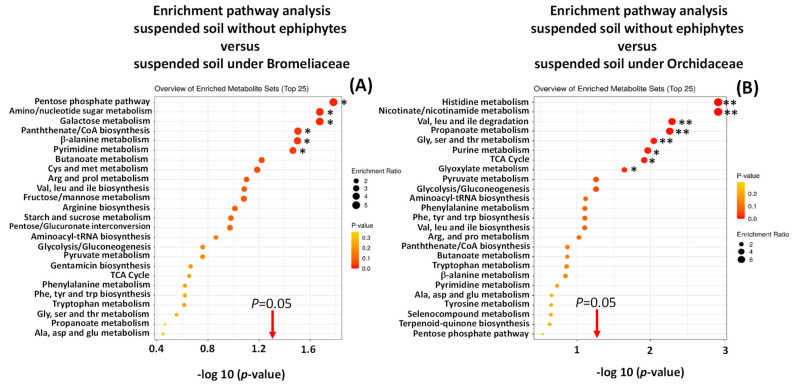
Enrichment pathway analysis of metabolomic profiles of suspended soils associated with Bromeliaceae (**A**) and Orchidaceae (**B**) compared with non-epiphyte soils. * *p* < 0.05, ** *p* < 0.01.

**Table 1 metabolites-11-00718-t001:** Differences in soil metabolome by soil type (forest floor, suspended soil +/− epiphyte) (*n* = 3) and tree species (*n* = 13), as tested by PERMANOVA.

	df	Sum of Squares	Mean Square	F	R^2^	P
Soil type	2	2.33	1.17	8.17	0.196	0.001
Tree species	12	2.61	0.22	1.52	0.219	0.001
Soil type × Tree species	13	1.83	0.14	0.98	0.153	0.56
Residuals	36	5.14	0.14	0.43	36	
Total	63	11.9	1			

**Table 2 metabolites-11-00718-t002:** Differences in soil metabolome by +/− epiphyte taxa and tree species (*n* = 13).

	df	Sum of Squares	Mean Square	F	R^2^	P
Epiphyte	4	3.47	0.50	4.00	0.291	0.001
Tree species	12	2.56	0.21	1.72	0.215	0.001
Epiphyte × Tree species	20	2.91	0.15	1.17	0.244	0.031
Residuals	24	2.98	0.12	0.25		
Total	63	11.9	1			

**Table 3 metabolites-11-00718-t003:** List of all collected vascular epiphytes of canopy soil attached to their base. The family to which the epiphyte belonged and, whenever possible, the genus and species are listed. Epiphytes are given by the identified tree species. We collected epiphytes from 13 out of the 14 sampled trees. *Asplenium* (*Asplenium serratum*), Bromeliaceae (*Aechmea aquilega* and *Mezobromelia pleiosticha*), Araceae (*Anthurium* sp. and *Philodendron* sp.), and Orchidaceae were sampled.

Tree Species	DBH	Epiphytes Collected	Epi Family	Epi Genus	Species
Aspidosperma sprucaneaum Benth. Ex Müll.Arg.	97.1	1	Bromeliaceae	*Achmea*	*A. aquilega*
Couratari oblongifolia Ducke & R. Knuth	79.1	1	Araceae	Unknown	*Philodendron* sp.
Eschweilera coriaceae (DC.) S.A.Mori	53.0	0	Araceae	Unknown	*Philodendron* sp.
Malvaceae Sterculia pruriens	53.6	3	Aspleniaceae	*Asplenium*	*Asplenium* sp.
			Bromeliaceae	Unknown	*M. pleiosticha*
			Araceae	*Philodendron*	*Philodendron* sp.
Annonaceae Oxandra asbeckii	53.2	2	Bromeliaceae	*Mezobromelia*	*M. pleiosticha*
			Orchidaceae	Unknown	Orchidaceae sp.
Lecythidaceae Gustavia hexapetala	48.7	1	Bromeliaceae	*Achmea*	*A. aquilega*
Chrysobalanaceae Licania alba	54.7	3	Aspleniaceae	*Asplenium*	*Asplenium* sp. 1
			Araceae	*Philodendron*	*Philodendron* sp.
			Bromeliaceae	*Achmea*	*A. aquilega*
Eperua falcata Aubl.	81.8	3	Aspleniaceae	*Asplenium*	*Asplenium* sp.
			Orchidaceae	Unknown	Orchidaceae sp. 2
			Araceae	*Philodendron*	*Philodendron* sp. *2*
Moraceae Brosimum guianense	73.5	2	Aspleniaceae	*Asplenium*	*Asplenium* sp. 1
			Orchidaceae	Unknown	Orchidaceae sp.
Lecythis persistens Sagot	47.4	3	Araceae	*Anthurium*	*Anthurium* sp. 2
			Bromeliaceae	Unknown	*M. pleiosticha*
			Orchidaceae	Unknown	Orchidaceae sp.
*Micropholis* sp.	69.6	2	Aspleniaceae	*Asplenium*	*Asplenium* sp.
			Bromeliaceae	*Achmea*	*A. aquilega*
Poraqueiba guianensis Aubl.	83.4	3	Orchidaceae	Unknown	Orchidaceae sp.
			Araceae	*Philodendron*	*Philodendron* sp.
			Araceae	*Philodendron*	*Philodendron* sp.
*Tetragastris* sp.	47.4	2	Araceae	*Philodendron*	*Philodendron* sp.
			Araceae	*Philodendron*	*Philodendron* sp.
Tetragastris altissima (Aubl.) Swart	95.5	3	Araceae	*Anthurium*	*Anthurium* sp.
			Aspleniaceae	*Asplenium*	*Asplenium* sp. 1
			Bromeliaceae	Unknown	*M. pleiosticha*

## Data Availability

All data is available from the corresponding author upon request. The data are not publicly available due to research still in progress.

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
