# Peer review of "Tree Species and Epiphyte Taxa Determine the “Metabolomic niche” of Canopy Suspended Soils in a Species-Rich Lowland Tropical Rainforest"

_metabolites, 2021, doi:10.3390/metabo11110718_

Round 1

Reviewer 1 Report

Dear authors,

I appreciate the clarity with which you have developed and written this paper based on a screening study in the French Guinea rainforest. I commend the presentation of results and writing style. The results are clearly stated. I find no typos or rewording changes. 

I would suggest a few points be addressed. In the methodology, did you prepare a composite sample of plant materials from several plants, rather than just one to minimise plant to plant variance? This is critical for making conclusions related to metabolomics and metabolite profiling, particularly if invoking software for pathways analysis. If not, how many samples were replicated. Generally three is insufficient to suitably profile. We typically work with 6 to 10 replicates and perform a process to create composite samples based on 3 plant minimum. 

Please elaborate more clearly how this was performed. Did you use GPS coordinates to delineate plants geographically or were they all collected on transects? How far away were they from each other?

Can you please list ionization conditions used. Can you also indicate how you addressed the issue of polar vs non-polar metabolites? Did you vary the column choice or the gradient to capture all? Did you use positive and negative modes for all samples and was this data a part of the analyses?

More details needed here. What was your quality control set up for processing your samples......i.e. blanks and standards?

More details needed.

In the results section, I would be keen to see an in depth result for at least one or two species ........or average over all species ........showing top 25 to 50 most abundant metabolites annotated.....perhaps you could present this data in a supplemental section broken down by species or averaged over genera for example. How many standards did you have on hand for identification? If none, it would be best to describe your results as annotated metabolites rather than identified........and important to suggest that the total number of metabolites are  molecular features......rather than compounds perhaps (fragments of molecules would be part of your analytics). Just be very careful of your wording so the methodology is clear. Metabolomics Society has standards for reporting and publishing that you may wish to refer to as well regarding GLP and presentation of results. 

A heat map profiling the abundance of key metabolites annotated in all species (let's say top 25) or averaged over ephiphyte, forest floor soil etc might also be very nice to present in the supplemental section as well. I'd like to see key compounds present in terms of phenolics, amino acids, etc profiled if possible. 

Author Response

Barcelona, September 20th, 2021

Dear Prof.

Dear authors,

1-) I appreciate the clarity with which you have developed and written this paper based on a screening study in the French Guinea rainforest. I commend the presentation of results and writing style. The results are clearly stated. I find no typos or rewording changes. 

Response: We appreciate the positive assessment of our article and its clarity.

2-) I would suggest a few points be addressed. In the methodology, did you prepare a composite sample of plant materials from several plants, rather than just one to minimise plant to plant variance? This is critical for making conclusions related to metabolomics and metabolite profiling, particularly if invoking software for pathways analysis. If not, how many samples were replicated. Generally three is insufficient to suitably profile. We typically work with 6 to 10 replicates and perform a process to create composite samples based on 3 plant minimum. 

Response: Thanks for the comment. We apologize for the lack of detailed explanation on the sampling protocol of our previous version. Thirteen species (Table 1) of trees across a range of families within the study plots were selected at the Pararé Research Station, randomly and based on the presence of epiphytes and the feasibility of climbing. We collected 3 replicates of soil around the individual tree always in the same position, less than 1 m of distance from the trunk base. We have now added more detailed information about sampling protocol in the revised text. This part of the new version of the manuscript now reads:

Soil sampling and analysis

Thirteen species (Table 1) of trees across a range of families within the study plots were selected at the Pararé Research Station, randomly and based on the presence of epiphytes and the feasibility of climbing. Using ropes to climb the trees, we collected specimens of Asplenium (Asplenium serratum), Bromeliaceae (Aechmea aquilega and Mezobromelia pleiosticha), Araceae (Anthurium sp. and Philodendron sp.), and Orchidaceae epiphytes from branches, recording their height and orientation in the canopy. The epiphyte-associated and suspended soils were removed from supporting branches, placed into sealed bag and transported to the field camp.  We sampled at least three epiphytes and three organic soils in each individual tree. Organic matters were removed from the roots of Araceae and Orchidaceae and from the tanks of bromeliads and litter-trapping leaf of Asplenium. The soils were processed following the approach described by Gargallo-Garriga et al. (2014), where samples were passed through a 0.5 x 0.5-cm mesh and placed in a paper bag, for temporary storage at -80 °C on dry ice and additional liquid nitrogen, prior to transfer to the EcoFoG lab at Kourou, where they were freeze dried for 48 h (Alpha 1-2 LD, Christ, Osterode am Harz, Germany). Then, samples were ground to a fine powder and homogenized, using a pestle and mortar, and stored at -80 °C prior to analysis.  In addition, we collected 3 replicates of soil (forest floor soil) around the individual tree always in the same position, less than 1 m of distance from the trunk base.

3-) Please elaborate more clearly how this was performed. Did you use GPS coordinates to delineate plants geographically or were they all collected on transects? How far away were they from each other?

Response: They were collected randomly within the fully inventoried 1.5-hectare plot at the Pararé Research Station. The distance between the trees are at least 10 m.

4-) Can you please list ionization conditions used. Can you also indicate how you addressed the issue of polar vs non-polar metabolites? Did you vary the column choice or the gradient to capture all? Did you use positive and negative modes for all samples and was this data a part of the analyses?

Response: We apologize for the lack of clarity of our previous version. We have rewritten this part with more extended information including the use of positive and negative ionization methods. The extraction was conducted using a 1:1 methanol: H2O solution thus extracting a mix of polar and non-polar compounds with mostly polar compounds. Thus the text now reads:

“LC-MS chromatograms were obtained with a Dionex Ultimate 3000 HPLC system (Thermo Fisher Scientific/Dionex RSLC, Dionex, Waltham USA) coupled to an LTQ Orbitrap XL high-resolution mass spectrometer (Thermo Fisher Scientific, Waltham, USA) equipped with an HESI II (heated electrospray ionisation) source. Chromatography was performed on a reversed-phase C18 Hypersil gold column (150 × 2.1 mm, 3-µ particle size; Thermo Scientific, Waltham, USA) at 30 °C. The mobile phases consisted of acetonitrile (A) and water (0.1% acetic acid) (B). Both mobile phases were filtered and degassed for 10 min in an ultrasonic bath prior to use. The elution gradient, at a flow rate of 0.3 mL per minute, began at 10% A (90% B) and was maintained for 5 min, then to 10% B (90% A) for the next 20 min. The initial proportions (10% A and 90% B) were gradually recovered over the next 5 min, and the column was then washed and stabilised for 5 min before the next sample was injected. The injection volume of the samples was 5 µL. HESI was used for MS detection. All samples were injected twice, once with the ESI operating in negative ionisation mode (-H) and once in positive ionisation mode (+H). The Orbitrap mass spectrometer was operated in FTMS (Fourier Transform Mass Spectrometry) full-scan mode with a mass range of 50-1000 m/z and high-mass resolution (60 000). The resolution and sensitivity of the spectrometer were monitored by injecting a standard of caffeine after every 10 samples, and the resolution was further monitored with lock masses (phthalates). Blank samples were also analysed during the sequence. The assignment of the metabolites was based on the standards, with the retention time and mass of the assigned metabolites in both positive and negative ionisation modes (Table S1).

Table S1. Processing parameters of LC-MS chromatograms using MzMine 2.0 (Pluskal et al., 2010). Chromatogram represents by the total ion current (TIC).

5-) More details needed here. What was your quality control set up for processing your samples......i.e. blanks and standards?

Response: We apologize for the lack of clarity of our previous version. Please see the response 4.

6-) In the results section, I would be keen to see an in depth result for at least one or two species ........or average over all species ........showing top 25 to 50 most abundant metabolites annotated.....perhaps you could present this data in a supplemental section broken down by species or averaged over genera for example. How many standards did you have on hand for identification? If none, it would be best to describe your results as annotated metabolites rather than identified........and important to suggest that the total number of metabolites are molecular features......rather than compounds perhaps (fragments of molecules would be part of your analytics). Just be very careful of your wording so the methodology is clear. Metabolomics Society has standards for reporting and publishing that you may wish to refer to as well regarding GLP and presentation of results. 

Response: Sorry. We now added a new paragraph to explain all the data. We clarified that we are working with a homemade database.

1.4. Metabolite identification and quantification

We identified and determined metabolites based on comparison with our standard compound library (>200 compounds) and searching for each MS and MS/MS in KEGG and MASSBANK databases, following the approach described by Gargallo-Garriga et al. (2020). For those that could not be positively identified we made tentative identifications based on MS/MS spectra as annotated metabolites. As a result, we assigned the stochiometry of 2757 metabolic compounds, and based on the stoichiometric ratios of nitrogen and hydrogen to carbon, and of oxygen to carbon (Koch et al., 2007; Kellerman et al., 2014), we classified the compounds into four broad families according to plant secondary metabolism comprising highly-unsaturated polyphenols (192 compounds), polycyclic aromatics (142 compounds), aliphatics (1351 compounds), and phenolic compounds (280 compounds).

7-) A heat map profiling the abundance of key metabolites annotated in all species (let's say top 25) or averaged over ephiphyte, forest floor soil etc might also be very nice to present in the supplemental section as well. I'd like to see key compounds present in terms of phenolics, amino acids, etc profiled if possible. 

Response: Please notice that this information is mostly provided  in the metabolome pathways (Figure 4).

Reviewer 2 Report

The use of metabolomic approach is very promising in order to study ecosystem biodiversity. It provides new insight to understand the complex interactions between soil and plants. This very descriptive article examines the metabolites diversity in very specific soils: suspended canopy soils and the impact of plants diversity (trees and epiphyte species). Methods used are relevant (chemical analysis, metabolite identification, statistics) but the manuscript is very confuse in particular the sampling design. The discussion is somehow brief: few elements on the enrichment of metabolic pathways or the potential key factors of the observed differences between metabolites profiles(see specific comments).

Introduction

Global comment: Canopy soils are very specific soils, you should add some information on their specificity compared to forest soils. Especially, the fact that these soils contain much more organic matters than floor soils. Because this will necessarily have an impact on the amount of metabolites present in these suspended soils.

Line 121:

Materials and methods

Line 138: Soils samples were collected from the base of each sample tree (forest floors?)?

Line 149: The description of the sampling design should be rewritten. For instance, you should explain the difference between “organic matter” and suspended soil, because in your discussion you compare these two systems as if they were two soils when they have, I think, very different characteristics. Likewise, “epiphyte-associated and suspended soil” correspond to epiphyte-associated suspended soil and suspended soil not associated with epiphyte? Be careful its quite confusing and it does not facilitate the reading of your article (forest floor…. I supposed it’s soil sampled at the base of the tree). you should add a table summarizing all the samples you have analyzed with a more explicit name (acronyms?)

Results

Line 224 (Table 2): its quite confusing….

Table 2 line 1: you compare the difference in soil metabolome of all forest floor/soil Vs all suspended soil associated or not with epiphyte?

Table 2 line 2: you compare the difference in soil metabolome between each tree species (forest floor or suspended soil, with or without associated epiphyte or by mixing all the soils…..)? Same question for table 3 line 2.

Line 228: Abundance of 90% of the metabolites were higher in suspended soils, why didn't you discuss this result (in line with my comment for the introduction).

Line 235: “690 compounds were greater in soils associated with epiphytes than those without (figure 1)” I cannot make the link between your sentence and the figure, because in the figure you compare "ephytes" (I presume suspended soils associated with an epiphyte) “organic matters” (it’s not a soil, it’s the organic matter remove from roots see M&M) and Forest floor soil (pay attention to the name once forest floor once forest floor soil). Where are the points corresponding to suspended soil non-associated with epiphyte?

Figure 1: Be careful, in the M&M part you did not precise that organic matter correspond to the non associated suspend soil. What is the color code for each point? All the other dots correspond to non-identified compounds?

Figure 2: What do the 13 points correspond to? the suspended soil metabolome (associated or not to epiphyte) or for each tree a mix of forest floor and suspended soil?

Figure 3: Epiphytes  correspond to metabolomic profiles of four epiphytes species whatever the tree and organic matters to metabolomic profile of all the suspended soils non associated with an epiphytes mix together?

Discussion

Global comment: in line with my previous comment for the introduction, to find more organic compounds resulting from the metabolization of the leaves in a "suspended" soil compared to a "classic soil" which contains 6 times less organic carbon, is not quite surprising. Did you try to characterize the organic constituents of the leaves of each species of tree (C, N, P contents – aromatic compounds….)?

Line 322: You could compare the variations you observe with those observed in rhizospheric soil.

Line 354: Which figure / table do you rely on to assert “metabolomic profiles associated with Asplenium varied with tree species?

Author Response

Reviewer 2

The use of metabolomic approach is very promising in order to study ecosystem biodiversity. It provides new insight to understand the complex interactions between soil and plants. This very descriptive article examines the metabolites diversity in very specific soils: suspended canopy soils and the impact of plants diversity (trees and epiphyte species). Methods used are relevant (chemical analysis, metabolite identification, statistics) but the manuscript is very confuse in particular the sampling design. The discussion is somehow brief: few elements on the enrichment of metabolic pathways or the potential key factors of the observed differences between metabolites profiles(see specific comments).

Response: Thanks for the positive comments about the general interest of our results.

We have now tried to improve and clarify the text in response to the reviewer’s comments. See responses point-by-point below.

Introduction

8-) Global comment: Canopy soils are very specific soils, you should add some information on their specificity compared to forest soils. Especially, the fact that these soils contain much more organic matters than floor soils. Because this will necessarily have an impact on the amount of metabolites present in these suspended soils.

Line 121:

 Response: We have added more background information about the specificity of canopy soils and in general about the expected differences between them and forest soils. The new text now reads:

“Tree diversity has been shown to influence soil heterogeneity in tropical forests (Musila et al., 2005), where impacts of differences in litter species composition and their released leachate shaped forest floor traits (Schreeg et al., 2013; Osborne et al. 2020) and also suspended canopy soils (Cardelus et al., 2009). Thus, changes in soil traits and contrasting microbe community composition, nutrient status, and enzyme content of forest floor and suspended soils have been observed depending on tree species (Matson et al., 2015; Donald et al., 2017; Looby et al., 2020).  Canopy soils have higher concentration of organic matter than ground soils (Nadkarni et al., 2002; 2004). A recent study in tropical mountain rainforest in Costa Rica has observed that canopy soils harbor very different symbiotic and fungi community that ground soils and have much more enzymatic activity (Looby et al., 2020). Here, we use metabolomic analyses to test the hypotheses that (i) tropical forest epiphyte-associated soil metabolome profiles vary depending on the epiphyte taxa, (ii) suspended soil and epiphyte-associated soil metabolomes differ in their composition and (iii) that these differences are shaped by tree species creating a wide array of distinct metabolome niches. We expect that canopy soil metabolomes contrast with those of the forest floor, and that the differences will depend on organic matter acquisition strategies of epiphytes or of microbes, within-canopy niche position and microclimate conditions (Scheffers et al. 2014), and host tree species characteristics (Cardelus et al., 2009).”

Materials and methods

9-) Line 138: Soils samples were collected from the base of each sample tree (forest floors?)?

Response: We apologize for the lack of detailed explanation on the sampling protocol of our previous version. Thirteen species (Table 1) of trees across a range of families within the study plots were selected at the Pararé Research Station, randomly and based on the presence of epiphytes and the feasibility of climbing. We collected 3 replicates of soil around the individual tree always in the same position, less than 1 m of distance from the trunk base. We have now added more detailed information about sampling protocol in the revised text. This part of the new version of the manuscript now reads:

Soil sampling and analysis

Thirteen species (Table 1) of trees across a range of families within the study plots were selected at the Pararé Research Station, randomly and based on the presence of epiphytes and the feasibility of climbing.. Using ropes to climb the trees, we collected specimens of Asplenium (Asplenium serratum), Bromeliaceae (Aechmea aquilega and Mezobromelia pleiosticha), Araceae (Anthurium sp. and Philodendron sp.), and Orchidaceae epiphytes from branches, recording their height and orientation in the canopy. The epiphyte-associated and suspended soils were removed from supporting branches, placed into sealed bag and transported to the field camp.  We sampled at least three epiphytes and three organic soils in each individual tree. Organic matters were removed from the roots of Araceae and Orchidaceae and from the tanks of bromeliads and litter-trapping leaf of Asplenium. The soils were processed following the approach described by Gargallo-Garriga et al. (2014), where samples were passed through a 0.5 x 0.5-cm mesh and placed in a paper bag, for temporary storage at -80 °C on dry ice and additional liquid nitrogen, prior to transfer to the EcoFoG lab at Kourou, where they were freeze dried for 48 h (Alpha 1-2 LD, Christ, Osterode am Harz, Germany). Then, samples were ground to a fine powder and homogenized, using a pestle and mortar, and stored at -80 °C prior to analysis. In addition, we collected 3 replicates of soil (forest floor soil) around the individual tree always in the same position, less than 1 m of distance from the trunk base.

10-) Line 149: The description of the sampling design should be rewritten. For instance, you should explain the difference between “organic matter” and suspended soil, because in your discussion you compare these two systems as if they were two soils when they have, I think, very different characteristics. Likewise, “epiphyte-associated and suspended soil” correspond to epiphyte-associated suspended soil and suspended soil not associated with epiphyte? Be careful its quite confusing and it does not facilitate the reading of your article (forest floor…. I supposed it’s soil sampled at the base of the tree). you should add a table summarizing all the samples you have analyzed with a more explicit name (acronyms?)

 Response: We clarified on the text. See response 9. Now we have used consistently the next three different soils: "epiphytes" (suspended soils associated with an epiphyte) “organic matters” (it’s a soil, it’s the organic matter from branches of the tree) and Forest floor soil.

Results

11-) Line 224 (Table 2): its quite confusing….

Response: The sentence and the table 2 show the differences in soil metabolome depending of soil type (forest floor, suspended soil with epiphyte and suspended soil without epiphyte) and tree species (N=13) tested using a PERMANOVA.

12-) Table 2 line 1: you compare the difference in soil metabolome of all forest floor/soil Vs all suspended soil associated or not with epiphyte?

Response: Yes we compared these three different soils (forest floor soil, canopy soil without epiphytes and canopy soil with epiphytes.

13-) Table 2 line 2: you compare the difference in soil metabolome between each tree species (forest floor or suspended soil, with or without associated epiphyte or by mixing all the soils…..)? Same question for table 3 line 2.

Response: See response before.in Table 2, line 2, yes, we compared the 13 tree species and soil type (3 soil types without differentiate the epiphyte soil in function of the species, df=n-1=3-1=2). Whereas in Table 3 line 2 we analyze the data separating the soils taking into account the epiphyte taxa (5 different types of soil df =n-1=5-1=4).

14-) Line 228: Abundance of 90% of the metabolites were higher in suspended soils, why didn't you discuss this result (in line with my comment for the introduction).

Response: We have added more background information about the specificity of canopy soils and in general about the expected differences between them and forest soils as we mention on response 8. In the discussion we have also commented this result. It now reads:

“Our results showed that abundance of several plant metabolite compounds, such as primary metabolites and phenolic groups, were greater in canopy soils than in forest floor soils, indicating a greater proportion of compounds of plant origin. This is consistent with previous studies observing that the organic matter contents highly differ in floor soil (~6 % C, compared to canopy soil or suspended soil ~35%C) (Nadkarni et al. 2004). We found higher abundances of metabolites such as amino acids, nucleotides, and compounds related to energy metabolism in suspended soils compared to forest floor soils, indicating the role of tree leaf litter in the formation of suspended soils. Similarly, higher concentrations of host tree compounds and nutrients have been reported from canopy soils than from forest floor soils in a Costa Rican rainforest, where there was also variation in the two soil types among tree species (Cardelus et al., 2009). This is again consistent with the origin of soil in canopies, which is mostly a result of plant litter in contrast with the ground soil, which is mostly a result of bedrock weathering and leaching (Zabowski et al., 2014)”

15-) Line 235: “690 compounds were greater in soils associated with epiphytes than those without (figure 1)” I cannot make the link between your sentence and the figure, because in the figure you compare "ephytes" (I presume suspended soils associated with an epiphyte) “organic matters” (it’s not a soil, it’s the organic matter remove from roots see M&M) and Forest floor soil (pay attention to the name once forest floor once forest floor soil). Where are the points corresponding to suspended soil non-associated with epiphyte?

Response: Thanks. We clarified on the text. "epiphytes" (suspended soils associated with an epiphyte) “organic matters” (it’s a soil, it’s the organic matter from branches of the tree) and Forest floor soil.

16-) Figure 1: Be careful, in the M&M part you did not precise that organic matter correspond to the non associated suspend soil. What is the color code for each point? All the other dots correspond to non-identified compounds?

Response: We have now added a detailed caption (see here below) that clarifies that “Soils are indicated by different colours (Pink square, ephytes; Pink clear, organic matters (canopy soil without epiphyte); Purplue triangle, Forest floor soil).

It also clarifies that “The various metabolomic families are represented by colors: dark blue, sugars; green, amino acids; orange, compounds involved in the metabolism of amino acids and sugars; cyan, nucleotides; brown, phenolics and red, others.”

Yes, we have also clarified that: “All the other small grey dots correspond to non-identified compounds”

Figure 1. Partial least squares discriminant analysis biplots of metabolomic profiles of forest floor soils and epiphyte associated and non-associated suspended soils (organic matter). Depicted values are means (±95% CI).  A) Samples categorised scores (mean ± S.E.) by different soil types:  "epiphytes" (suspended soils associated with an epiphyte) “organic matters” (it’s a soil, it’s the organic matter from branches of the tree without epiphytes) and Forest floor soil.  Soils are indicated by different colours (Pink square, ephytes; Pink clear, organic matters (canopy soil without epiphyte); Purplue triangle, Forest floor soil). (B) b) Loadings of the metabolites; only those with loadings >0.5 are depicted. The various metabolomic families are represented by colors: dark blue, sugars; green, amino acids; orange, compounds involved in the metabolism of amino acids and sugars; cyan, nucleotides; brown, phenolics and red, others. Metabolites: arginine (Arg), asparagine (Asn), aspartic acid (Asp), glutamic acid (Glu), glutamine (Gln), isoleucine (Ile), lysine (Lys), leucine (Leu), methionine (Met), phenylalanine (Phe), serine (Ser), tryptophan (Trp), threonine (Thr), tyrosine (Tyr), valine (Val), adenine (Ade), adenosine (Ado), thymidine (TdR), chlorogenic acid (CGA), trans-caffeic acid (CafA), α-ketoglutaric acid (KG), citric acid (Cit), L-malic acid (Mal), lactic acid (Lac), abscisic acid (Abs), pyruvate (Pyr), succinic acid (SAD), pantothenic acid hemicalcium salt (Pan), jasmonic acid (JA), 5,7-dihydroxy-3,4,5–trimethoxyflavone (Fla), acacetin (AC), epicatechin (EC), epigallocatechin (EGC), homoorientin (Hom), isovitexin (Ivx), kaempferol (Kae), myricetin (Myr), quercetin (Qct), resveratrol (Rvt), saponarin (Sp), catechin hydrate (Cat), 3-coumaric acid (CouA), gallic acid (GA), quinic acid (QuiA), sodium salicylate (Sal), syringic acid (Syr), trans-ferulic acid (Fer), vanillic acid (Van), 2-deoxy-D-ribose (Rib), D-(-)-lyxose (Lyx), D-(+)-sorbose (Sor), D-(+)-trehalose dehydrate (Tre) and aucubin (Auc). All the other small grey dots correspond to non-identified compounds

17-)Figure 2: What do the 13 points correspond to? the suspended soil metabolome (associated or not to epiphyte) or for each tree a mix of forest floor and suspended soil?

Response: The 13 points correspond to the forest floor soil of the 13 tree species. This is now indicated in the legend of the Figure 2.

 Figure 2. Partial least squares discriminant analysis biplots of metabolomic profiles of forest floor soil across 13 species of tree (Table 3). Depicted values are means (±95% CI).

18-) Figure 3: Epiphytes correspond to metabolomic profiles of four epiphytes species whatever the tree and organic matters to metabolomic profile of all the suspended soils non associated with an epiphytes mix together?

 Response: Sorry for the confusion. This soil is just the canopy soil under epiphytes.

Figure 3. Partial least squares discriminant analysis biplots of metabolomic profiles of epiphyte (Araceae, Asplenium, Bromeliaceae, Orchidae) associated and non-associated (organic matter) suspended soils. Depicted values are means (±95% CI).

Discussion

19-) Global comment: in line with my previous comment for the introduction, to find more organic compounds resulting from the metabolization of the leaves in a "suspended" soil compared to a "classic soil" which contains 6 times less organic carbon, is not quite surprising. Did you try to characterize the organic constituents of the leaves of each species of tree (C, N, P contents – aromatic compounds….)?

Response: Thanks for the comment. We have incorporated it in the Discussion by stating precisely what you mention here: “As expected, we found more organic compounds resulting from the metabolization of the leaves in canopy soils than in the forest floor soils which contain much less organic carbon, which is also completely consistent with the soil in canopies being the result of mostly of plant litter accumulation whereas ground soil is mostly result of bedrock weathering and leaching (Zabowski et al., 2014)”

20-) Line 322: You could compare the variations you observe with those observed in rhizospheric soil.

Response: Here in this part of the discussion we are discussing the results of previous studies showing a great contribution of epiphyte root exudates in the canopy soil composition, as an additional factor potentially contributing to the higher organic compound concentrations found in canopy soils mainly when they are associated to epiphytes.

21-) Line 354: Which figure / table do you rely on to assert “metabolomic profiles associated with Asplenium varied with tree species?,

Response: To represent the metabolomic profiles we use a new figure. It is a heat map profiling the abundance of key metabolites annotated in all epiphytes species.

Round 2

Reviewer 2 Report

Authors improve the quality of the manuscript and take into account our comments and answer our questions. I just have one suggestion. 

Line 280 (legend Figure 1): « organic matters » details in parentheses are not well written. i suggest you use what you write line 282.